# Factors associated with post-neonatal mortality in Ethiopia: Using the 2019 Ethiopia mini demographic and health survey

**Kenaw Derebe Fentaw** ⬤*, **Setegn Muche Fenta** ⬤, **Hailegebrael Birhan Biresaw**, **Mequanint Melkam Yalew**

Department of Statistics, Debre Tabor University, Debre Tabor, Ethiopia

* kenawzemen@gmail.com

**Editor:** Negar Rezaei, Non-Communicable Diseases Research Center, Endocrinology and Metabolism Population Sciences Institute, Tehran University of Medical Sciences, ISLAMIC REPUBLIC OF IRAN

## Abstract

### Background

Post-neonatal mortality is the number of deaths of infants aged 28 days through 11 months and is expressed as post-neonatal deaths per 1000 live births per year. This study aimed to identify the factors that influence post-neonatal death using the 2019 Ethiopia mini demographic and health survey (EMDHS2019).

### Methods

The study included 2126 post neonates born from mothers who had been interviewed about births in the five years before the survey. The survey gathering period was carried out from March 21, 2019, to June 28, 2019. The data were first analyzed with a chi-square test of association, and then relevant factors were evaluated with binary logistic regression models and the results were interpreted using adjusted odds ratio (AOR) and confidence interval (CI) of parameters.

### Results

The prevalence of post neonatal death was 16% (95% CI: 15.46, 17.78). The study also showed that not vaccinated post-neonates (AOR = 2.325, 95% CI: 1.784, 3.029), mothers who were not receiving any tetanus injection (AOR = 2.891, 95% CI: 2.254, 3.708), mothers age group 15-24(AOR = 1.836, 95% CI: 1.168, 2.886), Afar (AOR = 2.868, 95% CI: 1.264, 6.506), Somali(AOR = 2.273, 95% CI: 1.029, 5.020), Southern Nations, Nationalities, and People's Region(SNNP) (AOR = 2.619, 95% CI: 1.096, 6.257), 2–4 birth orders (AOR = 1.936, 95% CI: 1.225, 3.060), not attend antenatal care(ANC) visit (AOR = 6.491, 95% CI: 3.928, 10.726), and preceding birth interval less than 24 months (AOR = 1.755, 95% CI: 1.363,2.261) statistically associated with a higher risk of post neonatal death. Although not given anything other than breast milk (AOR = 0.604, 95% CI 0.462, 0.789), urban residents (AOR = 0.545, 95% CI: 0.338, 0.877), single births (AOR = 0.150, 95% CI: 0.096, 0.234), less than 3 children in a family (AOR = 0.665, 95% CI 0.470, 0.939) and the head of the male household (AOR = 0.442, 95% CI: 0.270, 0.724) were statistically associated with a lower risk of post-neonatal mortality.

**Data Availability Statement:** All relevant data are within the paper and its Supporting information files.

**Funding:** The author(s) received no specific funding for this work.

**Competing interests:** The authors have declared that no competing interests exist.

**Abbreviations:** ANC, Antenatal care; AOR, adjusted odds ratio and Health Survey; EAs, enumeration areas; EMDHS, Ethiopian Mini Demographic; PNC, postnatal care.

## Conclusions

It is highly suggested that maternal and child health care services (including antenatal care visits, postnatal care visits, and immunization) be strengthened and monitored during the early stages of infancy. Mothers from Somali, Afar, and SNNP regions, as well as multiple births, rural residents, and those giving birth to a child with a birth gap of fewer than 24 months, demand special care.

## Background

The useful and inexpensive indicator of population health is infant mortality. Infant mortality represents not only the health of the newborn but also the general well-being of society [1]. Post-neonatal mortality is the number of deaths of infants aged 28 days through 11 months and is expressed as post-neonatal deaths per 1000 live births in a year [2]. The prevalence of negative social, economic, and environmental conditions throughout the first year of life is reflected in the high postnatal mortality rate [3]. Infants' immune systems are substantially weaker than those of adults, making them much more sensitive to environmental and social problems. Furthermore, they are unable to care for themselves and must rely on others. As a result, children are usually the group that suffers the most from bad living conditions [4].

The infant mortality rate is very high in less developed countries as compared to developed countries [5, 6]. Children are more likely to die in their first month of life (neonatal), with an average global rate of 17 deaths per 1,000 live births in 2020, down from 37 deaths per 1,000 in 1990. In 2020, the risk of dying after the first month and before reaching the age of one (post-neonatal) was predicted to be 11 deaths per 1,000, while the risk of dying after reaching the age of one year and before reaching the age of five years (infant) was estimated to be 9 deaths per 1,000. Post neonatal deaths only slightly decrease each year in the world [7, 8]. However, some developing countries like Ethiopia are still far behind [9, 10].

The risk of post-neonatal mortality in African countries is 55 per 1000 live births, and this is more than five times higher compared to European countries where the rate is 10 per 1000 live birth [11, 12]. In Ethiopia, one in every 17 children dies before their first birthday [13]. The Ethiopian Demographic and Health Survey (EDHS), which was conducted in 2000, 2005, and 2011, revealed a significant decrease in newborn mortality [14]. The study carried out in different parts of the world [6, 15–20] revealed that the combined effect of birth order, the preceding birth interval, sex of the child, the maternal age at birth, the working status, parental education, the breastfeeding status, Mother's age, household wealth, religion, family size, were the important determinants associated with the risk of child mortality. These studies contributed to our understanding of many aspects of determinants of infant mortality in the world specifically, in Ethiopia. However, as far as our knowledge there is no study done on the post-neonatal mortality rate in Ethiopia using the binary logistic regression model. Furthermore, the previous study in Ethiopia was limited to small-scale survey data or focused more on descriptive statistical analyzes. Therefore, the present study focuses on investigating the factors associated with post-neonatal mortality in Ethiopia using the 2019 Ethiopia mini demographic and health survey (EMDHS2019).

## Methods

### Study area and data source

In the EMDHS, a community-based cross-sectional study was carried out from March 21, 2019, to June 28, 2019. Nine regional states (Afar, Tigray, Amhara, Oromia, Somali, Southern

Nations, Nationalities, and People's Region (SNNPR), Benishangul Gumuz, Gambella, and Harari) and two city administrations (Addis Ababa and Dire Dawa) are found in the country.

The sampling frame used for the 2019 EMDHS is a frame of all census enumeration areas created for the 2019 Ethiopian population and housing census conducted by the Central Statistical Agency (CSA). The survey used a two-stage stratified sampling technique. Each region was stratified into urban and rural areas, yielding 21 sampling strata. Samples of enumeration areas (EA) were selected independently in each stratum in two stages. A total of 305 EAs (212 in rural areas and 93 in urban areas) were chosen in the first stage, with probability proportional to EA size and independent selection in each sampling stratum. A household listing operation was carried out for all selected EAs. The generated list of households was used as a sampling frame for the second stage's selection of households. In the second step of the selection process, a specific number of 30 households in each group were chosen with an equal likelihood of systematic selection. A detailed methodology has been presented in the 2019 EMDHS final report [21]. In this study, a total weighted sample of 2126 post-neonates from mothers who were interviewed about births in the preceding 5 years before the survey was included in this analysis for this study Fig 1.

**Inclusion and exclusion criteria.** In the study, infants with complete data between the ages of 28 days and 11 months were included. Children older than 1 year and less than 28 days are not included in this study. Additionally, our analysis excludes post-neonates whose events (alive or dead) were missing (Fig 1).

**Variables of the study.** *Dependent variables*. The outcome variable of this study was post-neonatal death in months. Post-neonatal mortality is the number of deaths of infants aged 28 days through 11 months and is expressed as post-neonatal deaths per 1000 live births in a year. The outcome variable is coded as (0 = death and 1 = alive).

*Independent variables*. The Expected explanatory variables that were included in this study are socioeconomic, demographic, health, and environmental-related factors (Table 1).

**Data management and analysis.** After permission was granted for a reasonable request that explained the purpose of our study, the data were retrieved from the MEASURE DHS

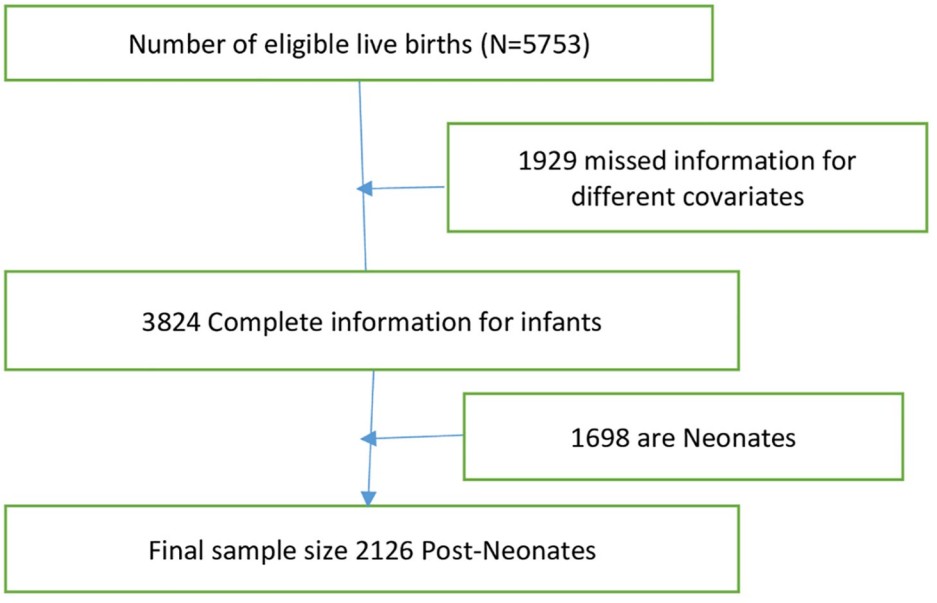

**Fig 1. The way of selecting a sample in the study.**

**Table 1. Coding and description of explanatory variables.**

| No | Variable | Description | Code |
|---|---|---|---|
| 1 | Region | Region of the household | 1 = Tigy |
| | | | 2 = Afar |
| | | | 3 = Amhara |
| | | | 4 = Oromia |
| | | | 5 = Somali |
| | | | 6 = Benishangul-Gumuz |
| | | | 7 = SNNP |
| | | | 8 = Gambela |
| | | | 9 = Harari |
| | | | 10 = Addis Abeba |
| | | | 11 = Dire Dawa |
| 2 | Place of residence | Type of place of residence | 1 = Urban, 2 = Rural |
| 3 | Source of water | Type of source for drinking water | 0 = protected,1 = unprotected |
| 4 | Toilet facility | Types of toilet facility | 0 = no use,1 = use toilet facility |
| 5 | Mother's Education | Mother's years of education | 0 = No education 1 = Primary |
| | | | 2 = secondary 3 = Higher and above |
| 6 | Place of delivery | Place of delivery | 0 = Home |
| | | | 1 = Health Sector |
| 7 | Number of ANC visit | Number of ANC visit | 0 = No ANC visit |
| | | | 1 = 1–4 |
| | | | 2 = greater than 4 |
| 8 | Tetanus Injection | Tetanus Injection during pregnancy | 0 = No |
| | | | 1 = Yes |
| 9 | Child Vaccinated | Child ever Vaccinated | 0 = No |
| | | | 1 = Yes |
| 10 | Wealth index | Wealth index of household | 0 = Poor |
| | | | 1 = Medium 2 = Rich |
| 11 | Sex | Sex of child | 1 = Male,2 = Female |
| 12 | Birth order | Birth order of child | 0 = first order |
| | | | 1 = 2–4 |
| | | | 2 = greater than 4 |
| 13 | Marital status | Marital status of the mother | 1 = Married |
| | | | 2 = Others |
| 14 | Age at first birth | Mothers age at first birth | 0 = less than 16 |
| | | | 1 = 17–32 |
| | | | 2 = 33–49 |
| The current | The current age of the mother | The current age of the mother | 0 = 15–24 |
| | | | 1 = 25–34 |
| | | | 2 = 35–49 |
| 16 | Duration of breastfeeding | Duration of breastfeeding | 0 = Never breastfed |
| | | | 1 = Still breastfeeding |
| | | | 2 = Others |
| 17 | Type of birth | Number of children that have a mother at a time of single delivery | 0 = Single birth |
| | | | 1 = Multiple birth |
| 18 | Preceding birth interval | Preceding birth interval (months) | 0 = less than or equal to 24 |
| | | | 1 = greater than 24 |

*(Continued)*

**Table 1.** (Continued)

| No | Variable | Description | Code |
|----|----------|-------------|------|
| 19 | Household head | Head of the household | 1 = male |
|    |          |             | 2 = female |
| 20 | No of children | Number of living children | 0 = less than or equal to 3 |
|    |          |             | 1 = greater than 3 |
| 21 | Given anything other than breast milk | Give a child anything other than breast milk | 0 = No |
|    |          |             | 1 = Yes |

program's official database (https://www.dhsprogram.com/data). The data from the 2019 EMDHS are open to all registered users. We extracted the response variable post-neonatal death and potential predictor variables after downloading the data. SPSS version 21 was used to extract our data set. After extracting data, Statistical analysis was performed. To describe the study participants, descriptive statistics such as frequencies, %ages, and charts were used. This study also used a combination of the chi-square test to determine whether the response variable was associated with different cofactors. Moreover, a multivariable binary logistic regression model must correspond to a response variable with two categories (post-neonatal mortality with two categories of yes or no).

A binary logistic regression model was used to determine the factors associated to post neonatal mortality. The outcome of the risk factor was reported in terms of an adjusted rating ratio with a significance level of 5% (95% CI). In the univariate analysis, a significance level of 25% was considered a candidate for the multivariate analysis of data analysis. All variables with p values $\leq 0.05$ were considered statistically significant.

For a binary response $Y_i$ and a quantitative explanatory variable $X_{ij}$, j = 1, 2. . . M and I = 1, 2. . . N, let $\pi_i = P(X_{ij})$ denote the "success probability" when $X_{ij}$ takes the values $X_{ij}$. The problem with the linear model is that the probability model E(Y) is used to approximate a probability value $\pi i = P(Y_i = 1)$ within the intervals 0 and 1, while $E(Y_i)$ Is not constrained. Therefore, we apply the logit transformation where the transformed quantity lies in the interval from minus infinity to positive infinity and is modeled as logit $(\pi_i) = \log \left(\frac{\pi i}{1-\pi i}\right) = \alpha + \beta_1 X_1 + \beta_2 X_2 + \cdots, +\beta_P X_P$

$\beta_i$ = the coefficient of the $i^{th}$ predictor variable determines the rate of increase or decrease of $X_{ij}$ On the log of the odds $Y_i = 1$, controlling for the other X's [22].

**Ethical considerations.** The study used secondary data analysis of publicly available survey data from the DHS program, and ethical approval and participant consent were not necessary for this study. We requested the DHS program, and permission was granted to download and use the data for this study from http://www.dhsprogram.com. There is no name of individuals or household addresses in the data files. Therefore, ethical approval was not necessary for this study.

## Results

### Socio-demographic characteristics

A total of 2126 post neonates took part in the study. Post-neonatal deaths occurred in 339 cases (16%). The highest proportions of post-neonatal deaths were found in Somali and Benishagul regional states (2.7%) and (2.1%) respectively. Rural residents had the highest %age of post-neonatal mortality (12.6%). Furthermore, half of those polled are uneducated (58.8%), and 60% of those polled are low-income (poor). More than half of the women (57.8%) did not

**Table 2. Sociodemographic characteristics of post neonatal mortality (EMDHS2019).**

| | | Child is alive | |
|---|---|---|---|
| | | No | Yes |
| Variables | Categories | Frequency (%) | Frequency (%) |
| Age in 5-year groups | 15–24 | 104(4.9) | 507(23.8) |
| | 25–34 | 144(6.8) | 934(43.9) |
| | 35–49 | 91(4.3) | 346(16.3) |
| Region | Tigray | 14(0.7) | 112(5.3) |
| | Afar | 35(1.6) | 288(13.5) |
| | Amhara | 23(1.1) | 140(6.6) |
| | Oromia | 41(1.9) | 240(11.3) |
| | Somali | 57(2.7) | 242(11.4) |
| | Benishangul | 45(2.1) | 154(7.2) |
| | SNNPR | 27(1.3) | 220(10.3) |
| | Gambela | 35(1.6) | 128(6) |
| | Harari | 29(1.4) | 119(5.6) |
| | Addis Adaba | 6(0.3) | 36(1.7) |
| | Dire Dawa | 27(1.3) | 108(5.1) |
| Type of place of residence | Urban | 72(3.4) | 292(13.7) |
| | Rural | 267(12.6) | 1495(70.3) |
| Highest educational level | No education | 191(9.0) | 1059(49.8) |
| | Primary | 122(5.7) | 546(25.7) |
| | Secondary | 17(0.8) | 129(6.1) |
| | Higher | 9(0.4) | 53(2.5) |
| Source of drinking water | Protected | 218(10.3) | 1010(47.5) |
| | Unprotected | 121(5.7) | 777(36.5) |
| Type of toilet facility | No | 148(7) | 915(43) |
| | Yes | 191(9) | 872(41) |
| Wealth index combined | Poor | 193(9.1) | 1084(51) |
| | Middle | 45(2.1) | 251(11.8) |
| | Rich | 101(4.8) | 452(21.3) |
| | Married | 310(14.6) | 1670(78.6) |
| | Others | 29(1.4) | 117(5.5) |
| Sex of household head | Male | 299(14.1) | 1428(67.2) |
| | Female | 40(1.9) | 359(16.9) |

have access to a safe/protectedly of drinking water and half of those polled used a toilet facility (Table 2).

## Obstetric characteristics

Mothers in the first birth order had the lowest rate of post-neonatal death (4%). The proportion of male and female post neonates was (9.2% and 6.8%) respectively. The majority (11%) of post-neonatal deaths were attributed to women who didn't receive tetanus injections during pregnancy. Furthermore, most mothers (38%) said they only had 1–4 prenatal checks while pregnant. In another regard, the majority of women (60.2%) gave birth at home and the majority of children (46.9%) were breastfed by their mothers. The highest proportion of deaths (12%) occurs in post-neonates who have never been immunized. Furthermore, all covariates are presented in (Table 3).

**Table 3. Obstetric characteristics of post neonatal mortality (EMDHS2019).**

| | | Child is alive | |
| | | No | Yes |
| Variables | Categories | Frequency (%) | Frequency (%) |
|---|---|---|---|
| of tetanus injections before birth | No | 235(11.1) | 784(36.9) |
| | Yes | 104(4.9) | 1003(47.2) |
| Give a child anything other than breast milk | No | 247(11.6) | 1459(68.6) |
| | Yes | 92(4.3) | 328(15.4) |
| Ever had vaccination | No | 256(12) | 1019(47.9) |
| | Yes | 83(3.9) | 768(36.1) |
| Place of delivery | Home | 197(9.3) | 1082(50.9) |
| | Health Sector | 142(6.7) | 705(33.2) |
| Age of mother at first birth | less than 16 | 111(5.2) | 593(27.9) |
| | 17–32 | 222(10.4) | 1187(55.8) |
| | 33–49 | 6(0.3) | 7(0.3) |
| Birth order number | First-order | 86(4) | 354(16.7) |
| | 2–4 | 119(5.6) | 790(37.2) |
| | greater than 4 | 134(6.3) | 643(30.2) |
| Type of birth | Single birth | 295(13.9) | 1748(82.2) |
| | multiple births | 44(2.1) | 39(1.8) |
| Sex of child | Male | 195(9.2) | 937(44.1) |
| | Female | 144(6.8) | 850(40) |
| Number of antenatal visits during pregnancy | No antenatal visits | 79(3.7) | 572(26.9) |
| | 1–4 | 69(3.2) | 740(34.8) |
| | greater than 4 | 191(9) | 475(22.3) |
| Preceding birth interval (months) | < = 24 | 111(5.2) | 388(18.3) |
| | > 24 | 228(10.7) | 1399(65.8) |
| Duration of breastfeeding | Never breastfed | 146(6.9) | 79(3.7) |
| | Still breastfeeding | 10(0.4) | 988(46.5) |
| | Others | 193(9.1) | 710(33.4) |
| Number of living children | less than or equal to 3 | 194(9.1) | 907(42.7) |
| | greater than 3 | 145(6.8) | 880(41.4) |

## Results of the binary logistic regression model

Binary logistic regression models were fitted utilizing categorical predictor variables that were found to be significant in the bivariate analysis using the enter selection (Likelihood ratio) approach. Table 4 summarizes the findings. Tetanus injection during pregnancy, giving the child anything other than breast milk, child vaccination, age of mother, region, birth order, type of birth, number of ANC visits, preceding birth interval, duration of breastfeeding, number of children, and head of household had statistically significant associations with post-neonatal deaths. When compared to vaccinated post neonates, the odds of post neonatal death were 2.325 times (AOR = 2.325, 95% CI: 1.784, 3.029) higher among those who had never been vaccinated. Mothers who did not receive tetanus injections during pregnancy had a 2.891 (AOR = 2.891, 95% CI: 2.254, 3.708) times higher risk of death after delivery than mothers who received any tetanus treatment during pregnancy. When comparing post neonates who were not given anything other than breast milk to post neonates who were given anything other than breast milk, the odds of post neonatal death were 0.604 times (AOR = 0.604, 95% CI:0.462,0.789) lower. When comparing women aged 15–24 to mothers aged 35–49, the risks

**Table 4. Logistic regression model for post neonatal death in Ethiopia (EMDHS2019).**

| Variables(Categories) | B | S.E. | Sig. | Exp(B) | 95% C.I.for EXP(B) | |
|---|---|---|---|---|---|---|
| | | | | | Lower | Upper |
| Tetanus injection during pregnancy(No) | 1.062 | .127 | .000 | 2.891 | 2.254 | 3.708 |
| Give the child anything other than breast milk(No) | -0.505 | 0.137 | .000 | 0.604 | 0.462 | 0.789 |
| Child vaccinated(No) | 0.844 | 0.135 | .000 | 2.325 | 1.784 | 3.029 |
| Age of mother(15–24) | 0.608 | 0.231 | .008 | 1.836 | 1.168 | 2.886 |
| Age of mother(25–34) | 0.043 | 0.309 | .890 | 1.044 | 0.569 | 1.915 |
| Region(Tigray) | 0.472 | 0.548 | .389 | 1.603 | 0.548 | 4.691 |
| Region(Afar) | 1.054 | 0.418 | .012 | 2.868 | 1.264 | 6.506 |
| Region(Amhara) | -0.492 | 0.497 | .322 | 0.611 | 0.231 | 1.620 |
| Region(Oromia) | 0.274 | 0.418 | .512 | 1.316 | 0.579 | 2.988 |
| Region(Somali) | 0.821 | 0.404 | .042 | 2.273 | 1.029 | 5.020 |
| Region(Benishangul) | -0.517 | 0.443 | .243 | 0.596 | 0.251 | 1.420 |
| Region(SNNPR) | 0.963 | 0.444 | .030 | 2.619 | 1.096 | 6.257 |
| Region(Gambela) | -0.044 | 0.463 | .924 | 0.957 | 0.386 | 2.371 |
| Region(Harari) | 0.559 | 0.444 | .208 | 1.749 | 0.733 | 4.173 |
| Region(Addis Ababa) | 0.834 | 0.677 | .218 | 2.303 | 0.611 | 8.685 |
| Residence(Urban) | -0.607 | 0.243 | .013 | 0.545 | 0.338 | 0.877 |
| BORD(First order) | 0.065 | 0.332 | .846 | 1.067 | 0.557 | 2.043 |
| BORD(2–4) | 0.661 | 0.234 | .005 | 1.936 | 1.225 | 3.060 |
| Type of birth(Single birth) | -1.90 | 0.229 | .001 | 0.150 | 0.096 | 0.234 |
| Number of ANC visits (No) | 1.870 | 0.256 | .000 | 6.491 | 3.928 | 10.726 |
| Number of ANC visits (1–4) | 0.929 | 0.223 | .000 | 2.533 | 1.636 | 3.923 |
| Preceding birth interval(less than 24 months) | 0.563 | 0.129 | .005 | 1.755 | 1.363 | 2.261 |
| Duration of breastfeeding(Never) | 1.917 | 0.162 | .000 | 6.799 | 4.954 | 9.331 |
| Duration of breastfeeding (Still feeding) | 8.093 | 46.6 | .286 | 2980.8 | 0.002 | 0.103 |
| No of children(less than 3) | -0.409 | 0.176 | .021 | 0.665 | 0.470 | 0.939 |
| Head of household(Male) | -0.817 | 0.252 | .001 | 0.442 | 0.270 | 0.724 |

Sig = significant (p<0.05), Exp(B) = exponential of B (odds ratio), S.E = standard error and CI = confidence interval.

of post neonatal death were 1.836 (AOR = 1.836, 95% CI 1.168, 2.886) times greater. When compared to post neonates in Dire Dawa city administrations, post neonates in Afar 2.868 (AOR = 2.868, 95% CI 1.264, 6.506), Somali 2.273 (AOR = 2.273, 95% CI 1.029, 5.020), and SNNP 2.619 (AOR = 2.619, 95% CI 1.096, 6.257) regions were more likely to die. When compared to rural people, the odds of post neonatal death in urban residents were 0.545 (AOR = 0.545, 95% CI 0.338, 0.877). Children born in the 2–4 birth order had 1.936 (AOR = 1.936, 95% CI: 1.225, 3.060) times greater chances of post neonatal mortality than those born in the five and above birth orders. The odds of post-neonatal death among singletons were 0.150 (AOR = 0.150, 95% CI: 0.096, 0.234) times lower as compared to multiple births. Compared to mothers who had five or more ANC visits, the risk of post-neonatal mortality was 6.491 times higher (AOR = 6.491, 95% CI: 3.928, 10.726). Furthermore, the death of post-neonatal children for mothers with 1–4 ANC visits was 2.533 times higher (AOR = 2.533, 95% CI: 1.636, 3.923) than for mothers with five or more ANC visits. Children born within 24 months of the previous birth interval had a 1.755 (AOR = 1.755, 95% CI 1.363–2.261) times greater risk of post-neonatal death than those born within 25 months. Those who never breastfed had a 6.799 (AOR = 6.799, 95% CI: 4.9549, 331) times higher risk of post-neonatal

death compared to other post-neonates. In a household with fewer than three children, the odds of post-neonatal death were 0.665 (AOR = 0.665, 95% CI, 0.470, 0.939) times greater than in a family with four or more children. The head of household was also a significant element in this study. The heads had a 0.442(AOR = 0.442, 95% CI: 0.270, 0.724) times lower risk of post-neonatal death than female household heads (Table 4).

### Assessment of goodness of fit of the model

After fitting the logistic model to categorical data, it is necessary to assess the suitability, adequacy, and utility of the model. To deal with this, we have a couple of options. The most often used approaches include Pearson's Chi-square, likelihood ratio tests (LRT), and Hosmer and Lemeshow Goodness of Fit tests. Based on the results in Table 5, the null hypothesis that there is no difference between the model with just a constant and the model with independent variables has been rejected. Because the r-square statistic for logistic regression models cannot be obtained precisely, these approximations are used instead. To a maximum of 1, larger pseudo-r-square values imply that the model can explain more of the variation. The statistical values of Cox & Snell (Pseudo R-square = 0.37) and Nagelkerke (Pseudo R-square = 0.641) were reasonable in this investigation, indicating that the model explained part of the variation in Table 6. Table 7 shows that the Hosmer-Lemeshow goodness of fit test does not yield a significant result. As a result, the model fits the data.

## Discussion

This study aims to identify determinants of post-neonatal death using data from the EMDHS2019. To investigate factors that affect post-neonatal mortality, both descriptive and binary logistic regression models were used. The descriptive statistics showed that the magnitude of post neonatal mortality was 16%. The binary logistic regression model revealed that tetanus injection during pregnancy, giving a child anything other than breast milk, child vaccinated, age of mother, region, birth order, type of birth, number of ANC visits, preceding birth interval, duration of breastfeeding, number of children, and head of household had statistically significant associations with post neonatal deaths.

A major determinant of post-neonatal death was the number of ANC visits. The risk of newborn death is significantly reduced as the frequency of ANC visits increases. This study's findings are consistent with those of previous studies [23–28]. This outcome could be explained by the fact that ANC visits are required to improve the health of mothers and fetuses by lowering pregnancy complications. Long birth intervals had a lower risk of post-neonatal death than short birth intervals, and the risk of post-neonatal death decreased as the prior birth interval grew. The risk of obstetric complications is higher in mothers with short birth intervals than in mothers with longer birth intervals [29]. In Ethiopia, the order of birth had a huge impact on post-neonatal death. With an increase in the post-neonatal birth order, the probability of post-neonatal mortality increased. As the mother's child care responsibilities grow, the amount of child care provided is expected to decrease. This finding corroborated previous research findings [30, 31].

**Table 5. Omnibus tests of model coefficients.**

|  | Chi-square | Df | Sig. |
|---|---|---|---|
| Step | 5.419 | 1 | .002 |
| Block | 995.759 | 26 | .000 |
| Model | 995.759 | 26 |  |

**Table 6. Model summary.**

| Step | -2 Log-likelihood | Cox & Snell R Square | Nagelkerke R Square |
|---|---|---|---|
| | 869.866[a] | .374 | .640 |

Giving a child anything other than breast milk was found to be the cause of post-neonatal mortality. The findings corroborate previous studies [32, 33]. Breastfeeding is a child's first line of defense against mortality and disease, protecting them from respiratory infections, gastrointestinal illnesses, and other negative health consequences [34, 35]. The female household head was linked to a higher rate of post-neonatal mortality. This finding was consistent with a study from the year before [36–38]. This is because female-headed households are more likely to face food insecurity and are less likely to be properly vaccinated [39]. As a result, they are vulnerable to vaccine-preventable sickness and death. This shows that female-headed families will have difficulty making decisions on post-neonatal health because they will be preoccupied with other domestic and social responsibilities. The probability of post-neonatal death increased with each unit increase in the age of the household head. This could be because as people get older, the likelihood of providing child care decreases.

When compared to mothers who did not receive a tetanus shot during pregnancy, mothers who received a tetanus shot were less likely to lose their babies during the post-neonatal period. The outcome is in line with expectations [40–42]. This could be because tetanus vaccination produces protective antibodies against post-neonatal tetanus. The study found that post neonates born in large families have a much higher risk of post neonatal mortality than those born in small families. As a result, the risk of post-neonatal mortality increases in tandem with the size of the family. Because large households are more likely to share facilities, this is the case. Several studies corroborate our findings [43–45]. Post neonates who were breastfed by their mother had a lower chance of death than those who were never breastfed. This could be explained by the fact that nursing protects babies from infectious disorders since breast milk is high in antibodies and white blood cells. This outcome is consistent with previous findings [24, 26, 46]. The type of birth was found to be a statistically significant predictor of postnatal death. The risk of post-neonatal death was higher in multiple births than in single births. Multiple births have a lower weight competition due to food consumption [47]. This result is also similar to [48]. Vaccinated post-neonates have a decreased risk of death than non-vaccinated post-neonates, which is consistent with earlier research findings [43, 47].

The mother's age was found to be a significant predictor of post-neonates' death. Post-neonatal death was higher in the younger mother (early age) than in the older mother. This outcome is consistent with earlier findings [49–51]. Physical and physiological immaturity, as well as a higher likelihood of insufficient weight gain during pregnancy, are plausible explanations for this finding [52]. The findings also revealed that the location of birth was a significant risk factor for post-neonatal mortality. The risk of post neonatal death was greater in rural neonates than in urban neonates. This is because newborns in urban areas have greater access to health care and other key health-related amenities that are required for neonatal survival [29]. These studies also agree with the previous study [42, 53]. Furthermore, statistically, geographical regions were linked to post-neonatal death. Compared to the Dire dawa city administration,

**Table 7. Hosmer and Lemeshow test.**

| Chi-square | Df | Sig. |
|---|---|---|
| 4.440 | 8 | .815 |

mothers from the Somali, Afar, and SNNP regions had a higher incidence of post neonatal death. The implementation of effective health policies varies by location, which could be the cause of this geographical variation. This is similar to prior studies [54–56].

This study finding have important policy implications, especially in determining the program needs for a sustainable decline in post-neonatal mortality rate, and in monitoring public health interventions. It is important to look beyond identifying associated factor with death of post-neonates. Increasing mother's education and empowerment may help reduce childhood mortality. Reducing motherhood in younger ages and increasing the spacing between births are also necessary to reduce post-neonatal deaths. Some other important contextual factors such as quality and care of health facility, cultural practices, customs, environmental conditions, etc. could not be addressed in this study due to the unavailability of data in the DHS. The authors suggest further studies considering these unobserved factors that are likely to be associated with post-neonatal mortality to better understand the association between family and community level factors and post-neonatal mortality in Ethiopia. Interventions and strategies should be targeted focusing on these characteristics to improve child health outcomes as well as the future betterment of Ethiopia.

## Conclusions

The result of this study showed that Tetanus injection during pregnancy, giving a child anything other than breast milk, child vaccination, age of mother, region, birth order, type of birth, number of ANC visits, preceding birth interval, duration of breastfeeding, number of children, and head of household had statistically significant associations with post neonatal mortality. It is highly suggested that maternal and child health care services (including ANC, PNC, and immunization) be strengthened and monitored during the early stages of infancy. Multiple births, rural inhabitants, and giving birth to a kid with a birth gap of less than 24 months, demand special care. We recommend also health institutions made effort to give awareness to mothers about contraceptive use and breastfeeding to reduce post-neonatal mortality.

Policies and programs that aim to address regional differences in post-neonatal mortality must be vigorously developed and implemented. Special attention must be paid to the Afar, Somali, and SNNP regions to achieve this.

## Supporting information

**S1 Data.**
(SAV)

## Acknowledgments

The authors would like to thank the measure DHS program for granting access to the EMDHS data sets.

## Author Contributions

**Data curation:** Kenaw Derebe Fentaw, Mequanint Melkam Yalew.

**Formal analysis:** Kenaw Derebe Fentaw.

**Methodology:** Setegn Muche Fenta, Mequanint Melkam Yalew.

**Software:** Kenaw Derebe Fentaw, Setegn Muche Fenta.

**Supervision:** Setegn Muche Fenta, Hailegebrael Birhan Biresaw.

**Validation:** Setegn Muche Fenta, Hailegebrael Birhan Biresaw.

**Visualization:** Hailegebrael Birhan Biresaw.

**Writing – original draft:** Kenaw Derebe Fentaw, Mequanint Melkam Yalew.

**Writing – review & editing:** Hailegebrael Birhan Biresaw.

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
