## [Decision Letter · Decision Letter 0]

19 May 2022

PONE-D-22-07946Factors associated with Post-Neonatal mortality in Ethiopia: Using 2019 Ethiopia mini demographic and health surveyPLOS ONE

Dear Dr. Fentaw,

Thank you for submitting your manuscript to PLOS ONE. After careful consideration, we feel that it has merit but does not fully meet PLOS ONE’s publication criteria as it currently stands. Therefore, we invite you to submit a revised version of the manuscript that addresses the points raised during the review process.

There are several major and minor issues regarding this manuscript, and it is not suitable for publication in its current form. It is strongly recommended general language editing and revisions regarding grammar and syntax by a native speaker or an expert since there are several grammatical errors and typos in the text.The PLOS Data policy requires authors to make all data underlying the findings described in their manuscript fully available without restriction, with rare exception (please refer to the Data Availability Statement in the manuscript PDF file). The data should be provided as part of the manuscript or its supporting information, or deposited to a public repository. For example, in addition to summary statistics, the data points behind means, medians and variance measures should be available. If there are restrictions on publicly sharing data—e.g. participant privacy or use of data from a third party—those must be specified.

We look forward to receiving your revised manuscript.

Kind regards,

Negar Rezaei, M.D., Ph.D.,

Academic Editor

PLOS ONE

a) Did participants provide their written or verbal informed consent to participate in this study?

Reviewers' comments:

Reviewer's Responses to Questions

**Comments to the Author**

1. Is the manuscript technically sound, and do the data support the conclusions?

Reviewer #1: Yes

Reviewer #2: Partly

Reviewer #3: Yes

2. Has the statistical analysis been performed appropriately and rigorously? 

Reviewer #1: Yes

Reviewer #2: Yes

Reviewer #3: Yes

3. Have the authors made all data underlying the findings in their manuscript fully available?

Reviewer #1: Yes

Reviewer #2: No

Reviewer #3: Yes

4. Is the manuscript presented in an intelligible fashion and written in standard English?

Reviewer #1: Yes

Reviewer #2: No

Reviewer #3: No

5. Review Comments to the Author

Reviewer #1: Thank you for the opportunity to review the study entitled " Factors associated with Post-Neonatal mortality in Ethiopia: Using 2019 Ethiopia mini demographic and health survey”.

The honorable authors have discussed the infant mortality as a sensitive indicator of population health. The current study is noteworthy owing two facts: 1. Post-neonatal mortality is an essential index for community health in all societies 2. Since some developing countries have not reached their goal for providing required infrastructures, this study could have some information for policymakers to implement at the society level. The introduction part has emphasized the potential advantages of the current study well. The methods and results have comprehensively described the methods and findings. From an overall point of view, binary logistic regression has some limitations, although due to the overall design of study, applying other methods of analysis is not possible. The paper is well written, and the Discussion section has comprehensively described the observed results, although attention to the following issues could improve the quality of the paper:

Comments:

1. The article has lots of punctuation issues (like in abstract section, lines 11,12), which should be corrected in the whole manuscript.

2. As the keywords of the study, the authors should provide more keywords. Also, Binary Logistic Regression is not a good keyword in this regard.

3. The whole instruction part is one paragraph, which might be hard for the reader to follow the clues of the study. The introduction should be divided into two to three paragraphs.

4. The discussion part of the study has described the findings well. However, one of the essential content of this study is its application at the society level. The authors should add two sections at the end of the discussion section. First, they should provide comprehensive data from countries with successful strategies that could effectively control infant mortality. So, these strategies could be used for developing countries. Second, the authors should provide a paragraph for policymakers. This should contain some strategies and concepts for policymakers to be implemented at society level.

5. Providing DOI for references is highly encouraged.

Reviewer #2: Thank you for giving me the opportunity to review this paper. It was a cross-sectional investigation in Ethiopia on factors associated with post-neonatal mortality using 2019 Ethiopia mini demographic and health survey data. It is an important public health issue in less developed regions of the world, and lessons learned from such studies could be valuable for other similar countries. However, there are several major and minor issues regarding this manuscript, and it is not suitable for publication in its current form. The draft needs thorough and careful major revisions to provide a more suitable manuscript.

Major concerns:

General language editing and revisions regarding grammar and syntax are highly recommended by a native speaker or an expert since there are several grammatical errors and typos in the text.

In the abstract section, the authors have mentioned that “The goal of this study was to identify the factors that influence post neonatal death using EMDHS2019.” The EMDHS2019 survey needs to be defined and explained in the methods of the abstract. The methods section in the abstract is vague and needs more details and descriptions.

As mentioned in the methods section of the study, 2126 “post neonates” were included in the final analyses. However, in the methods of the abstract section, it has been mentioned as 2126 “women aged 15 to 49.” Please make them identical and provide more explicit methods. Based on the results, it seems that “women” were studied, not specifically the post neonates.

I wonder why the women were analyzed in this study!! It would be much better to perform analyses based on the number of the post neonates. i.e., of the total 2126 included women, how many births did they have? Then, you could examine how many of the total birth numbers in the included population died during the post-neonatal period. Please revise the methods and the results in this regard to provide better estimates. For example, “n” number of born children were assessed in the included 2126 women, of which “n” percent passed away in the post-neonatal period instead of mentioning 16% of women experienced post-neonatal deaths among their children. In the first paragraph of the discussion section, the authors have also mentioned that “the magnitude of post-neonatal mortality was 16%.” However, it is not an accurate interpretation since women were assessed, not children. All these parts need major revisions.

In figure 1, it has been captured that 5753 women were eligible. Please describe in the methods how these women were selected and what were the inclusion and exclusion criteria.

Regarding the data source, it is mentioned in some parts that EMDHS data were assessed. In the methods section, it has been mentioned that EMDHS was a secondary source of data. This section lacks an explicit and comprehensive description of the main data source and the means of data collection. Please provide more explanations around the data collection process and inclusion and exclusion criteria.

Please provide research ethics related to this study in the methods section. Did consent forms were obtained from the subjects?

In methods, Dependent variables: “The outcome variable of this study was post neonatal death in months. The outcome variable coded as (0=death and 1=alive)”. Please explain death in “how many” months after birth was defined as post-neonatal death in this study. You need to vividly define the post-neonatal death period, for example, deaths from birth to “n” months after birth.

In the statistical analysis section, please specify and name the tests that were utilized. Were they logistic regression analyses? Please mention them clearly in the methods section.

In the binary logistic regression section of the results, in the sentence “When comparing children born with single births to children born with multiple births, the odds of post-neonatal death were.150 (AOR =.150, 95 percent CI: .096, .234) times lower.” What do you mean by “single births and multiple births”? Please define them and make them clear.

Minor issues:

In the first line of the background section, instead of “the most sensitive indicator,” please mention “a useful or important indicator…” as the cited reference 1 also concluded that it is a useful and inexpensive indicator, not necessarily “the most sensitive” one.

Background lines 13,14; please define the age of one and five. Are they “years”? Besides, “infancy” does not apply to the age of five years, which has been written in parentheses. Please reword this sentence to provide a more explicit one for the readers. In the following sentence, “Post neonatal only slightly…,” please mention “post-neonatal deaths.”

In the last sentences of the background section, please mention “our” knowledge instead of my knowledge.

It is redundant to provide the expanded term for EMDHS each time in the manuscript. As you wrote it down once in the background, please write down the abbreviation afterward in the methods and other sections.

In the binary logistic regression section of the results, correct the values such as .545 to 0.545 in the text. Please edit all the values and provide the correct form.

Reviewer #3: This paper aimed to identify the factors that influence postneonatal death using the 2019 Ethiopia mini demographic and health survey. As a developing country in Africa, Ethiopia needs more consideration from health policies to decrease the burden of diseases. In developing countries maternal and neonatal diseases are still imposing a considerable burden. Therefore, considering related studies to investigate the ongoing neonatal and maternal diseases is critical. The study carries out important messages, but to make the paper suitable to be published some issues should be addressed. The comments are as follows.

Major issues:

The paper needs some editing in English fluency and the draft should become more professional. As an example: “However, as far as my knowledge” my determiners should be changed to our, because authors are more than one person. Similar situations can be found in other parts of the draft.

Table 4 has results basically copied from all presented by SPSS software, I suggest just using practical results in this table with the format that is popular with other published papers.

Minor issues:

In the abstract, some abbreviations were used without defining them, which possibly most readers are not familiar with them.

The background part of the abstract can be reduced to two sentences because, except for the aim sentence, the information presented here is not necessary for the abstract part.

Please add the inclusion and exclusion criteria to the methods part.

If in this study all the respondents are mothers, I suggest replacing “respondents” with “mothers” in the manuscript.

The ANC and PNC visits were mentioned in the discussion which is very important. I suggest discussing more the numbers of this study and comparing it with other studies and other similar countries

6. PLOS authors have the option to publish the peer review history of their article (what does this mean?). If published, this will include your full peer review and any attached files.

Reviewer #1: No

Reviewer #2: No

Reviewer #3: No

---

## [Author Response · Author response to Decision Letter 0]

30 May 2022

Authors' responses to the comments of the Editors and Reviewers

To begin, I am grateful for your thoughtful review and evaluation of this study, Factors Associated with Post-Neonatal Mortality in Ethiopia: Using the 2019 Ethiopia Mini Demographic and Health Survey. We revised our manuscript in response to the reviewers' comments and suggestions. The following is a point-by-point response to the remarks and suggestions of reviewers and editors.

We trust that we have satisfactorily addressed them and that the manuscript is now ready for publishing.

Response to academic editor suggestions and comments

Comment1: Please ensure that your manuscript meets PLOS ONE's style requirements, including those for file naming.

Response1: We appreciate your suggestion. The file names were changed to conform to the style criteria. We should now have no deviations from the style criteria.

Comment2: Please amend your current ethics statement to address the following concerns: Did participants provide their written or verbal informed consent to participate in this study?

Response2: Thank you for this suggestion. The study used secondary data analysis of publicly available survey data from the DHS program, ethical approval, and participant consent were not necessary for this study. We requested the DHS program, and permission was granted to download and use the data for this study from http://www.dhsprogram.com.Therefore, no need for written or verbal informed consent to participate in this study.

Comment3: In your Data Availability statement, you have not specified where the minimal data set underlying the results described in your manuscript can be found.

Response3: Thank you for this suggestion. The data in this study were secondary and are available in the public domain at (https://www.dhsprogram.com/data). If you need specific data(sample) used in this study, we will submit.

uploaded minimal data set as a Supporting Information file.

Comment4: There are several grammatical errors and typos in the text.

Response4: Thank you for your important comments. We repeatedly read the whole document and consider all the corrections and the language is edited by professionals.

Response to Reviewers’ comments and suggestions

 Dear reviewers, we are contented and appreciate your careful and thorough reading of this manuscript and we would like to say thank you for your thoughtful and constructive comments and suggestions. Based on your comments and suggestions, we carefully corrected the manuscript and addressed every comment one by one. We made rephrasing in some parts of the paragraphs of the manuscript accordingly.

Reviewer #1

 Comment1: The article has lots of punctuation issues (like in the abstract section, line 11,12), which should be corrected in the whole manuscript.

Response1: Thank you dear reviewer for your important comments. We repeatedly read the whole document and consider all the corrections.

Comment2: As the keywords of the study, the authors should provide more keywords. Also, Binary Logistic Regression is not a good keyword in this regard.

Response2: Thank you, dear reviewer, for improving our manuscript. In the revised manuscript, we have made changes based on your feedback.

Comment3: The whole instruction part is one paragraph, which might be hard for the reader to follow the clues of the study. The introduction should be divided into two to three paragraphs.

Response3: Thank you for your suggestions. Based on your recommendation we divided it into three paragraphs.

Comment4: The authors should add two sections at the end of the discussion section. First, they should provide comprehensive data from countries with successful strategies that could effectively control infant mortality. So, these strategies could be used in developing countries. Second, the authors should provide a paragraph for policymakers. This should contain some strategies and concepts for policymakers to be implemented at the society level. 

Response4: Thank you for your constructive comments. We have incorporated the comments in the revised manuscript at the end of discussion and conclusion parts.

Comment5: Providing DOI for references is highly encouraged.

Response5: Thank you for your important suggestion. We incorporated DOIs of each reference. 

Reviewer #2: 

Comment1: General language editing and revisions regarding grammar and syntax are highly recommended by a native speaker or an expert since there are several grammatical errors and typos in the text.

Response1: Thank you for your important comments. We repeatedly read the whole document and consider all the corrections and the language is edited by professionals.

Comment2: The methods section in the abstract is vague and needs more details and descriptions

Response2: Thank you dear reviewer for your constructive comments. We accepted your comments and gadetailedail explanation in the revised manuscript.

Comment3: As mentioned in the Methods section of the study, 2126 “post neonates” were included in the final analyses. However, in the methods of the abstract section, it has been mentioned as 2126 “women aged 15 to 49.” Please make them identical and provide more explicit methods.

Response3: We appreciate all of your insightful and critical feedback. It is a critical comment. We went over all of the errors in the manuscript. ETKR (child dataset) with 5753 infants was used. Dear Reviewer! If we were to use the ETIR dataset (women dataset), your suggestion would be the best method to proceed. However, we did not use a women's data set for this study.

Comment4: In figure 1, it has been captured that 5753 women were eligible. Please describe in the methods how these women were selected and what the inclusion and exclusion criteria were?

Response4: Thank you for your comments. First of all, the data was not women, those 5753 data were child datasets. So, we have made changes. We included inclusion and exclusion criteria in the revised manuscript.

Comment5: Please provide more explanations around the data collection process and inclusion and exclusion criteria.

Response5: Thank you for your recommendation. We made a change to the revised manuscript and included the points that you have raised.

Comment6: Please provide research ethics related to this study in the methods section. Did consent forms obtained from the subjects?

Response6: Thank you for your comments. Based on your request, we have included Ethical considerations in the methods section. Since, the study used secondary data analysis of publicly available survey data from the DHS program, ethical approval, and participant consent were not necessary for this study.

Comment7: Please explain death in “how many” months after birth was defined as post-neonatal death in the outcome variable.

Response7: Thank you for your constructive comments. We have put the specified period on the revised document.

Comment8: In the statistical analysis section, please specify and name the tests that were utilized. Were they logistic regression analyses? Please mention them clearly in the methods section

Response8: Thank you for these interesting comments. We described it in detail in the revised manuscript.

Comment9: When comparing children born with single births to children born with multiple births, the odds of post-neonatal death were.150 (AOR =.150, 95 percent CI: .096, .234) times lower.” What do you mean by “single births and multiple births”? Please define them and make them clear.

Response9: Thank you for your comments. We have made changes in the description of independent variables as well as in the result parts.

Comment 10: About Minor issues:

Response 10: Thank you dear reviewer for all your interesting comments and suggestions. We have read the whole document repeatedly and made revisions for all your minor issues.

Reviewer #3: 

Comment1: The paper needs some editing in English fluency and the draft should become more professional

Response 1: Thank you dear reviewer for your constructive comments and suggestions. We repeatedly read the whole document and consider all the corrections and the language is edited by professionals.

Comment2: Table 4 has results copied from all presented by SPSS software, I suggest just using practical results in this table with the format that is popular with other published papers.

Response 2: Thank you for your interesting comments. We have changed the format of all tables based on references from different published articles.

Comment3: About minor issues:

Response 3: Thank you for your insightful comments. The issues you identified are critical and have a big impact on the article's quality. We accepted everything and made changes to the updated document.

Sincerely, 

On behalf of all authors, 

Kenaw Derebe Fentaw

---

## [Decision Letter · Decision Letter 1]

28 Jun 2022

PONE-D-22-07946R1Factors associated with Post-Neonatal mortality in Ethiopia: Using 2019 Ethiopia mini demographic and health surveyPLOS ONE

Dear Dr. Fentaw,

Thank you for submitting your manuscript to PLOS ONE. After careful consideration, we feel that it has merit but does not fully meet PLOS ONE’s publication criteria as it currently stands. Therefore, we invite you to submit a revised version of the manuscript that addresses the points raised during the review process.

We look forward to receiving your revised manuscript.

Kind regards,

Negar Rezaei, M.D., Ph.D.,

Academic Editor

PLOS ONE

Journal Requirements:

Reviewers' comments:

Reviewer's Responses to Questions

**Comments to the Author**

1. If the authors have adequately addressed your comments raised in a previous round of review and you feel that this manuscript is now acceptable for publication, you may indicate that here to bypass the “Comments to the Author” section, enter your conflict of interest statement in the “Confidential to Editor” section, and submit your "Accept" recommendation.

Reviewer #1: All comments have been addressed

Reviewer #2: (No Response)

Reviewer #3: (No Response)

2. Is the manuscript technically sound, and do the data support the conclusions?

Reviewer #1: Yes

Reviewer #2: Yes

Reviewer #3: Yes

3. Has the statistical analysis been performed appropriately and rigorously? 

Reviewer #1: Yes

Reviewer #2: Yes

Reviewer #3: Yes

4. Have the authors made all data underlying the findings in their manuscript fully available?

Reviewer #1: Yes

Reviewer #2: Yes

Reviewer #3: No

5. Is the manuscript presented in an intelligible fashion and written in standard English?

Reviewer #1: Yes

Reviewer #2: Yes

Reviewer #3: Yes

6. Review Comments to the Author

Reviewer #1: Dear Authors,

I have read the revised version of the manuscript “Factors associated with Post-Neonatal mortality in Ethiopia: Using 2019 Ethiopia mini demographic and health survey”. Overall, all the comments have been addressed and the quality of the manuscript has been improved.

Reviewer #2: Thank you for providing a revised version of this manuscript. Some descriptions regarding data and analyzes were vague in the original version. The authors have reflected most of the comments and provided more explicit descriptions in the revised version.

Reviewer #3: This paper aimed to identify the factors that influence post-neonatal death using the 2019 Ethiopia mini demographic and health survey. The paper was revised once upon the reviewers' comments. I want to thank the authors for their efforts. But to make the draft in a scientifically acceptable form, I think some steps are ahead.

As the authors mentioned about the data availability "all data are fully available without restriction", I could not find the specific data regarding this study at https://www.dhsprogram.com/data web address. Please provide and upload the dataset you used to analyze.

Thanks to the authors for trying to improve the language and to remove grammatical errors, but I think just minimal revisions were done, and it can be improved more. Just to mention that in the authors' response to my comment about grammatical issues, I found grammatical issues!

As I commented about revising table 4, I think changes are not enough, because some statistical data are presented which are not necessarily within the scope of this study. For example, why the "degrees of freedom" is necessary to be presented here. Also, other labels are statistical abbreviations, to present it as a medical scientific article, the labels should be changed in this regard.

In the abstract part, CI, AOR, and ANC abbreviations are not defined yet.

A recommendation: the sentence "Post-neonatal mortality is the number of deaths of infants aged 28 days through 11 months and is expressed as post-neonatal deaths per 1000 live births per year." is too basic and is not necessary enough to be presented in the abstract.

The data gathering time period is not presented in the methods of the abstract part.

About the inclusion and exclusion criteria part: some wording issues should be corrected. For example, the sentence "Infants whose ages are under 28 days and greater than 1 year are excluded from this study." is technically wrong because infants are aged between 28 days and one year, and children younger or older are not considered as infants. The sentence "Also, missing information about the outcome variable was excluded from the study." is vague. to revise this part, first mention all included participants, then who are excluded and why you excluded them.

7. PLOS authors have the option to publish the peer review history of their article (what does this mean?). If published, this will include your full peer review and any attached files.

Reviewer #1: No

Reviewer #2: No

Reviewer #3: No

---

## [Author Response · Author response to Decision Letter 1]

1 Jul 2022

I attached line by line response for reviewer #3 comments.

---

## [Decision Letter · Decision Letter 2]

12 Jul 2022

Factors associated with Post-Neonatal mortality in Ethiopia: Using 2019 Ethiopia mini demographic and health survey

PONE-D-22-07946R2

Dear Dr. Fentaw,

We’re pleased to inform you that your manuscript has been judged scientifically suitable for publication and will be formally accepted for publication once it meets all outstanding technical requirements.

Kind regards,

Negar Rezaei, M.D., Ph.D.,

Academic Editor

PLOS ONE

Additional Editor Comments (optional):

Reviewers' comments:

Reviewer's Responses to Questions

**Comments to the Author**

1. If the authors have adequately addressed your comments raised in a previous round of review and you feel that this manuscript is now acceptable for publication, you may indicate that here to bypass the “Comments to the Author” section, enter your conflict of interest statement in the “Confidential to Editor” section, and submit your "Accept" recommendation.

Reviewer #3: All comments have been addressed

2. Is the manuscript technically sound, and do the data support the conclusions?

Reviewer #3: Yes

3. Has the statistical analysis been performed appropriately and rigorously? 

Reviewer #3: Yes

4. Have the authors made all data underlying the findings in their manuscript fully available?

Reviewer #3: Yes

5. Is the manuscript presented in an intelligible fashion and written in standard English?

Reviewer #3: Yes

6. Review Comments to the Author

Reviewer #3: I read the revised version of the manuscript entitled “Factors associated with Post-Neonatal mortality in Ethiopia: Using 2019 Ethiopia mini demographic and health survey”. I want to thank the authors for their tremendous efforts in revising the drafts. I think all my comments have been addressed, and the paper is in an acceptable format now.

7. PLOS authors have the option to publish the peer review history of their article (what does this mean?). If published, this will include your full peer review and any attached files.

Reviewer #3: No

---

## [Editor Report · Acceptance letter]

15 Jul 2022

PONE-D-22-07946R2 

Factors associated with Post-neonatal mortality in Ethiopia: Using the 2019 Ethiopia mini demographic and health survey 

Dear Dr. Fentaw:

I'm pleased to inform you that your manuscript has been deemed suitable for publication in PLOS ONE. Congratulations! Your manuscript is now with our production department. 

Kind regards, 

on behalf of

Dr. Negar Rezaei 

Academic Editor

PLOS ONE